# Analytical Study of Momentum-Based Acceleration Methods in Paradigmatic High-Dimensional Non-Convex Problems

**Stefano Sarao Mannelli**
Department of Experimental Psychology
University of Oxford
Oxford, United Kingdom
`stefano.saraomannelli@psy.ox.ac.uk`

**Pierfrancesco Urbani**
Université Paris-Saclay, CNRS, CEA
Institut de physique théorique
Gif-sur-Yvette, France
`pierfrancesco.urbani@ipht.fr`

## Abstract

The optimization step in many machine learning problems rarely relies on vanilla gradient descent but it is common practice to use momentum-based accelerated methods. Despite these algorithms being widely applied to arbitrary loss functions, their behaviour in generically non-convex, high dimensional landscapes is poorly understood. In this work, we use dynamical mean field theory techniques to describe analytically the average dynamics of these methods in a prototypical non-convex model: the (spiked) matrix-tensor model. We derive a closed set of equations that describe the behaviour of heavy-ball momentum and Nesterov acceleration in the infinite dimensional limit. By numerical integration of these equations, we observe that these methods speed up the dynamics but do not improve the algorithmic threshold with respect to gradient descent in the spiked model.

## 1 Introduction

In many computer science applications one of the critical steps is the minimization of a cost function. Apart from very few exceptions, the simplest way to approach the problem is by running local algorithms that move down in the cost landscape and hopefully approach a minimum at a small cost. The simplest algorithm of this kind is gradient descent, that has been used since the XIX century to address optimization problems [1]. Later on, faster and more stable algorithms have been developed: second order methods [2, 3, 4, 5, 6, 7] where information from the Hessian is used to adapt the descent to the local geometry of the cost landscape, and first order methods based on momentum [8, 9, 10, 11, 12] that introduce inertia in the algorithm and provably speed up convergence in a variety of convex problems. In the era of deep-learning and large datasets, the research has pushed towards memory efficient algorithms, in particular stochastic gradient descent that trades off computational and statistical efficiency [13, 14], and momentum-based methods are very used in practice [15]. Which algorithm is the best in practice seems not to have a simple answer and there are instances where a class of algorithms outperforms the other and vice-versa [16]. Most of the theoretical literature on momentum-based methods concerns convex problems [17, 18, 19, 20, 21] and, despite these methods have been successfully applied to a variety of problems, only recently high dimensional non-convex settings have been considered [22, 23, 24]. Furthermore, with few exceptions [25], the majority of these studies focus on *worst-case* analysis while empirically one could also be interested in the behaviour of such algorithms on typical instances of the optimization problem, formulated in terms of a generative model extracted from a probability distribution.

The main contribution of this paper is the analytical description of the average evolution of momentum-based methods in two simple non-convex, high-dimensional, optimization problems. First we consider the mixed $p$-spin model [26, 27], a paradigmatic random high-dimensional optimization problem.

Furthermore we consider its spiked version, the spiked matrix-tensor [28, 29] which is a prototype high-dimensional non-convex inference problem in which one wants to recover a signal hidden in the landscape. The second main result of the paper is the characterization of the algorithmic threshold for accelerated-methods in the inference setting and the finding that this seems to coincide with the threshold for gradient descent.

The definition of the model and the algorithms used are reported in section 2. In section 3 and 4 we use dynamical mean field theory [30, 31, 32] to derive a set of equations that describes the average behaviour of these algorithms starting from random initialization in the high dimensional limit and in a fully non-convex setting.

We apply our equations to the spiked matrix-tensor model [29, 33, 34], which displays a similar phenomenology as the one described in [24, 35] for the phase retrieval problem: all algorithms have two dynamical regimes. First, they navigate in the non-convex landscape and, second, if the signal to noise ratio is strong enough, the dynamics eventually enters in the basin of attraction of the signal and rapidly reaches the bottom of the cost function. We use the derived state evolution of the algorithms to determine their algorithmic threshold for signal recovery.

Finally, in Sec. 5 we show that in the analysed models, momentum-based methods only have an advantage in terms of speed but they do not outperform vanilla gradient descent in terms of the algorithmic recovery threshold.

## 2 Model definition

We consider two paradigmatic non-convex models: the mixed $p$-spin model [32, 36], and the spiked matrix-tensor model [28, 29]. Given a tensor $\boldsymbol{T} \in (\mathbb{R}^N)^{\otimes p}$ and a matrix $\boldsymbol{Y} \in \mathbb{R}^{N \times N}$, the goal is to find a common low-rank representation $\boldsymbol{x}$ that minimizes the loss

$$\mathcal{L} = -\frac{1}{\Delta_p} \sqrt{\frac{(p-1)!}{N^{p-1}}} \sum_{i_1,\dots,i_p=1}^{N} T_{i_1,\dots,i_p} x_{i_1} \dots x_{i_p} - \frac{1}{\Delta_2} \frac{1}{\sqrt{N}} \sum_{i,j=1}^{N} Y_{ij} x_i x_j, \tag{1}$$

with $\boldsymbol{x}$ in the $N$-dimensional sphere of radius $\sqrt{N}$. The two problems differ by the definition of the variables $\boldsymbol{T}$ and $\boldsymbol{Y}$. Call $\boldsymbol{\xi}^{(p)}$ and $\boldsymbol{\xi}^{(2)}$ order $p$ tensor and a matrix having i.i.d. Gaussian elements, with zero mean and variances $\Delta_p$ and $\Delta_2$ respectively. In the mixed $p$-spin model, tensor and matrix are completely random $\boldsymbol{T} = \boldsymbol{\xi}^{(p)}$ and $\boldsymbol{Y} = \boldsymbol{\xi}^{(2)}$. While in the spiked matrix-tensor model there is a low-rank representation given by $\boldsymbol{x}^* \in \mathcal{S}^{N-1}(\sqrt{N})$ embedded in the problem as follows:

$$T_{i_1 \dots i_p} = \sqrt{\frac{(p-1)!}{N^{p-1}}} x_{i_1}^* \dots x_{i_p}^* + \xi_{i_1 \dots i_p}^{(p)}, \qquad Y_{ij} = \frac{x_i^* x_j^*}{\sqrt{N}} + \xi_{ij}^{(2)}. \tag{2}$$

These problems have been studied both in physics, and computer science. In the physics literature, research has focused on the relationship of gradient descent and Langevin dynamics and the corresponding topology of the complex landscape [32, 37, 38, 36, 39, 27, 40]. The state evolution of the gradient descent dynamics for the mixed spiked matrix-tensor model has been studied only more recently [33, 34]. All these works considered simple gradient descent dynamics and its noisy (Langevin) dressing.

In this work we focus on accelerated methods and provide an analytical characterization of the average performance of these algorithms for the models introduced above. In order to simplify the analysis we relax the hard constraint on the norm of the vector $\boldsymbol{x}$ and consider $\boldsymbol{x} \in \mathbb{R}^N$ while adding a penalty term to $\mathcal{L}$ to enforce a soft constraint $\frac{\mu}{4N} \left( \sum_i x_i^2 - N \right)^2$, so that the total cost function is $\mathcal{H} = \mathcal{L} + \frac{\mu}{4N} \left( \sum_i x_i^2 - N \right)^2$. Using the techniques described in detail in the next section we write the state evolution for the following algorithms:

- **Nesterov acceleration** [9] starting from $\boldsymbol{y}[0] = \boldsymbol{x}[0] \in \mathbb{S}^{N-1} \left( \sqrt{N} \right)$

$$\boldsymbol{x}[t+1] = \boldsymbol{y}[t] - \alpha \nabla \mathcal{H}(\boldsymbol{y}[t]), \tag{3}$$

$$\boldsymbol{y}[t+1] = \boldsymbol{x}[t+1] + \frac{t}{t+3} \left( \boldsymbol{x}[t+1] - \boldsymbol{x}[t] \right). \tag{4}$$

given $\alpha$ the learning rate of the algorithm.

- **Polyak's or heavy ball momentum** (HB) [8] starting from $\boldsymbol{y}[0] = \boldsymbol{0}$ and $\boldsymbol{x}[0] \in \mathbb{S}^{N-1}\left(\sqrt{N}\right)$, given the parameters $\alpha, \beta$

$$\boldsymbol{y}[t+1] = \beta \boldsymbol{y}[t] + \nabla \mathcal{H}(\boldsymbol{x}[t]), \tag{5}$$
$$\boldsymbol{x}[t+1] = \boldsymbol{x}[t] - \alpha \boldsymbol{y}[t+1]; \tag{6}$$

- **gradient descent** (GD) starting from $\boldsymbol{x}[0] \in \mathbb{S}^{N-1}\left(\sqrt{N}\right)$

$$\boldsymbol{x}[t+1] = \boldsymbol{x}[t] - \alpha \nabla \mathcal{H}(\boldsymbol{x}[t]). \tag{7}$$

This case has been considered in [27, 33] with the constraint $\sum_i x_i^2 = N$. The generalization to the present case in which constraint is soft is a straightforward small extension of these previous works.

We will not compare the performance of these accelerated gradient methods to algorithms of different nature (such as for example message passing ones) in the same settings. Our goal will be the derivation of a set of dynamical equations describing the average evolution of such algorithms in the high dimensional limit $N \to \infty$.

## 3 Dynamical mean field theory

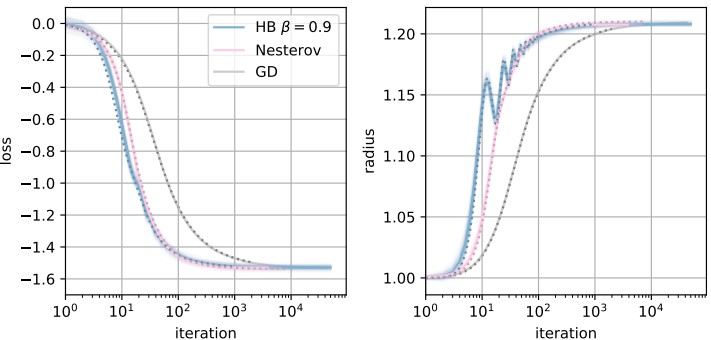

Figure 1: **Simulation and DMFT comparison in mixed $p$-spin model.** The simulations in the figures have parameters $p = 3$, $\Delta_3 = 2/p$, $\Delta_2 = 1$, ridge parameter $\mu = 10$ and input dimension $N = 1024$. In all our simulations we use the dilution technique [41, 42] to reduce the computational cost. We consider: Nesterov acceleration in pink; heavy ball momentum in blue with $\alpha = 0.01$ and $\beta = 0.9$; and gradient descent in grey. We run 100 simulations (in transparency) and draw the average. The parameters for heavy ball are the best parameters found in our simulations, see also Fig. 2 for a comparison. The results from the DMFT equations are drawn with dotted lines.

We use dynamical mean field theory (DMFT) techniques to derive a set of equation describing the evolution of the algorithms in the high-dimensional limit. The method has its origin in statistical physics and can be applied to the study of Langevin dynamics of disordered systems [30, 31, 43]. More recently it was proved to be rigorous in the case of the mixed $p$-spin model [44, 45]. The application to the inference version of the optimization problem is in [29, 33]. The same techniques have also been applied to study the stochastic gradient descent dynamics in single layer networks [46] and in the analysis of recurrent neural networks [47, 48, 49].

The derivation presented in the rest of the section is heuristic and, as such, it is not fully rigorous. Making our results rigorous would be an extension of the works [44, 45] where path-integral methods are used to prove a large deviation principle for the infinite-dimensional limit. Our non-rigorous results are checked against extensive numerical simulations.

The idea behind DMFT is that, if the input dimension $N$ is sufficiently large, one can obtain a description of the dynamics in terms of the typical evolution of a representative entry of the vector $\boldsymbol{x}$ (and vector $\boldsymbol{y}$ when it applies). The representative element evolves according to a non-Markovian

stochastic process whose memory term and noise source encode, in a self-consistent way, the interaction with all the other components of vector $\boldsymbol{x}$ (and $\boldsymbol{y}$). The memory terms as well as the statistical properties of the noise are described by dynamical order parameters which, in the present model, are given by the dynamical two-time correlation and response functions.

In this first step of the analysis we obtain an effective dynamics for a representative entry $x_i$ (and $y_i$). The next step consists in using such equations to compute self-consistently the properties of the corresponding stochastic processes, namely the memory kernel and the statistical correlation of the noise. In Fig. 1 we anticipate the results by comparing numerical simulations with the integration of the DMFT equations for the different algorithms: on the left we observe the evolution of the loss, on the right we observe the evolution of the radius of the vector $\boldsymbol{x}$, defined as the $L_2$ norm of the vector $||\boldsymbol{x}||_2$. We find a good agreement between the DMFT state evolution and the numerical simulations.

We compare Nesterov acceleration with the heavy ball momentum in the mixed $p$-spin model Fig. 1, and in the spiked model Fig. 3. Nesterov acceleration allows for a fast convergence to the asymptotic energy without need of parameter tuning. In Fig. 2 we compare the numerical simulations for the HB algorithm and the DMFT description of the corresponding massive momentum version for several control parameters.

**DMFT equations**

In the following we describe the resulting DMFT equations for the correlation and response functions. The details of their derivation for the case of the Nesterov acceleration are provided in the following section, while we leave the other cases to the supplementary material (SM). The dynamical order parameters appearing in the DMFT equations are one-time or two-time correlations, e.g. $C_{xy}[t,t'] = \sum_i x_i[t]y_i[t']/N$, and response to instantaneous perturbation of the dynamics, e.g. $R_x[t,t'] = \left(\sum_i \delta x_i[t]/\delta H_i[t']\right)/N$ by a local field $\boldsymbol{H}[t'] \in \mathbb{R}^N$ where the symbol $\delta$ denotes the functional derivative. In this section we show only the equations for the mixed $p$-spin model and we discuss the difference and the derivation of the equations for the spiked tensor in the SM.

From the order parameters we can evaluate useful quantities that describe the evolution of the algorithms. In particular in Figs. 1,2,3 we show the loss, the radius, and the overlap with the solution in the spiked case (Fig. 3):

- Average loss

$$\mathcal{L}[t] = -\frac{\alpha}{\Delta_p C_x[t,t]^{\frac{p}{2}}} \sum_{t''=0}^{t} R_x[t,t']C_x[t,t']^{p-1} - \frac{\alpha}{\Delta_2 C_x[t,t]} \sum_{t''=0}^{t} R_x[t,t']C_x[t,t']; \quad (8)$$

- Radius $\sqrt{C_x[t,t]}$;
- Define $m_x[t] = \frac{1}{N}\sum_i x_i[t]x_i^*$ an additional order parameter for the spiked matrix-tensor model (more details are given in the SM), the overlap with ground truth is

$$\frac{\boldsymbol{x}[t]\cdot\boldsymbol{x}^*}{||\boldsymbol{x}||} = \frac{m_x[t]}{\sqrt{C_x[t,t]}}$$

**Nesterov acceleration.** It has been shown that this algorithm has a quadratic convergence rate to the minimum in convex optimization problems under Lipschitz loss functions [9, 50], thus it outperforms standard gradient descent whose convergence is linear in the number of iterations. The analysis of the algorithm is described by the flow of the following dynamical correlation functions

$$C_x[t,t'] = \frac{1}{N}\sum_i x_i[t]x_i[t'], \quad (9)$$

$$C_y[t,t'] = \frac{1}{N}\sum_i y_i[t]y_i[t'], \quad (10)$$

$$C_{xy}[t,t'] = \frac{1}{N}\sum_i x_i[t]y_i[t'], \quad (11)$$

$$R_x[t,t'] = \frac{1}{N}\sum_i \frac{\delta x_i[t]}{\delta H_i[t']}, \tag{12}$$

$$R_y[t,t'] = \frac{1}{N}\sum_i \frac{\delta y_i[t]}{\delta H_i[t']}. \tag{13}$$

The dynamical equations are obtained following the procedure detailed in section 4. Call $Q(x) = x^2/(2\Delta_2) + x^p/(p\Delta_p)$,

$$C_x[t+1,t'] = C_{xy}[t,t'] - \alpha\mu\left(C_y[t,t]-1\right)C_y[t,t'] + \alpha^2\sum_{t''=0}^{t'} R_x[t',t'']Q'\left(C_y[t,t'']\right) + $$
$$+ \alpha^2\sum_{t''=0}^{t} R_y[t,t'']Q''\left(C_y[t,t'']\right)C_{xy}[t',t'']; \tag{14}$$

$$C_{xy}[t+1,t'] = C_y[t,t'] - \alpha\mu\left(C_y[t,t]-1\right)C_{xy}[t,t'] + \alpha^2\sum_{t''=0}^{t'} R_y[t',t'']Q'\left(C_y[t,t'']\right) + $$
$$+ \alpha^2\sum_{t''=0}^{t} R_y[t,t'']Q''\left(C_y[t,t'']\right)C_y[t',t'']; \tag{15}$$

$$C_{xy}[t',t+1] = \frac{2t+3}{t+3}C_x[t+1,t'] - \frac{t}{t+3}C_x[t,t']; \tag{16}$$

$$C_y[t',t+1] = \frac{2t+3}{t+3}C_{xy}[t+1,t'] - \frac{t}{t+3}C_{xy}[t,t']; \tag{17}$$

$$R_x[t+1,t'] = R_y[t,t'] + \delta_{t,t'} - \alpha\mu\left(C_y[t,t]-1\right)R_y[t,t']$$
$$+ \alpha^2\sum_{t''=t'}^{t} R_y[t,t'']R_y[t'',t']Q''\left(C_y[t,t'']\right); \tag{18}$$

$$R_y[t',t+1] = \frac{2t+3}{t+3}R_x[t+1,t'] - \frac{t}{t+3}R_x[t,t']. \tag{19}$$

The initial conditions are: $C_x[0,0] = 1$, $C_y[0,0] = 1$, $C_{xy}[0,0] = 1$, $R_x[t+1,t] = 1$, $R_y[t+1,t] = \frac{2t+3}{t+3}$.

The equations show a discretized version of the typical structure of DMFT equations. We can observe: terms immediately ascribable to the dynamical equations (3,4) and summations whose interpretation is less trivial without looking into the derivation. They represent memory kernels that take into account linear response theory for small perturbations to the dynamics (e.g. the last term of Eq. (14)) and a noise whose statistical properties encode the effect of all the degrees of freedom on a representative one (e.g. the second last term of Eq. (14)).

**Heavy ball momentum.** The DMFT equations are obtained analogously to previous ones,

$$C_y[t+1,t'] = \beta C_y[t,t'] + \mu\left(C_x[t,t]-1\right)C_{xy}[t,t'] + \alpha\sum_{t''=0}^{t'} R_y[t',t'']Q'\left(C_x[t,t'']\right)$$
$$+ \alpha\sum_{t''=0}^{t} R_x[t,t'']Q''\left(C_x[t,t'']\right)C_{xy}[t'',t']; \tag{20}$$

$$C_{xy}[t',t+1] = \beta C_{xy}[t',t] + \mu\left(C_x[t,t]-1\right)C_x[t,t'] + \alpha\sum_{t''=0}^{t'} R_x[t',t'']Q'\left(C_x[t,t'']\right)$$
$$+ \alpha\sum_{t''=0}^{t} R_x[t,t'']Q''\left(C_x[t,t'']\right)C_x[t',t'']; \tag{21}$$

$$C_{xy}[t+1,t'] = C_{xy}[t,t'] - \alpha C_y[t+1,t']; \tag{22}$$

$$C_x[t+1,t'] = C_x[t,t'] - \alpha C_{xy}[t',t+1]; \tag{23}$$

$$R_y[t+1,t'] = \beta R_y[t,t'] + \frac{1}{\alpha}\delta_{t,t'} + \mu\left(C_x[t,t]-1\right)R_x[t,t']$$

$$+ \alpha\sum_{t''=0}^{t} R_x[t,t'']R_x[t'',t']Q''\left(C_x[t,t'']\right); \tag{24}$$

$$R_x[t+1,t'] = R_x[t,t'] - \alpha R_y[t+1,t']. \tag{25}$$

with initial conditions: $C_x[0,0] = 1$, $C_y[0,0] = 0$, $C_{xy}[0,0] = 0$, $R_y[t+1,t] = 1/\alpha$, $R_x[t+1,t] = -1$. Fig. 2 shows the consistency of theory and simulations.

Mappings between discrete update equation and continuous flow for both heavy ball momentum and Nesterov acceleration have been proposed in the literature. In the SM we considered the work [51] that maps HB to second order ODEs in some regimes of $\alpha$ and $\beta$. This mapping establishes the equivalence of the algorithm to the physics problem of a massive particle moving under the action of a potential. This problem has been studied in [52] but the result is limited to the fully under-damped regime where there is no first order derivative term, corresponding therefore to a dynamics that is fully inertial and which never stops due to energy conservation. In the SM we obtain the dynamical equations for arbitrary damping regimes, and we recover the equivalence established in [51] comparing the results from the two DMFTs formulations.

**Gradient descent.** A simple way to obtain the gradient descent DMFT is by taking the limit $m \to 0$ in the DMFT of the massive momentum description of HB. We get

$$C_x[t+1,t'] = C_x[t,t'] - \alpha\mu\left(C_x[t,t]-1\right)C_x[t,t'] + \alpha^2\sum_{t''=0}^{t'} R_x[t',t'']Q'\left(C_x[t,t'']\right)$$

$$+ \alpha^2\sum_{t''=0}^{t} R_x[t,t'']Q''\left(C_x[t,t'']\right)C_x[t',t'']; \tag{26}$$

$$R_x[t+1,t'] = R_x[t,t'] + \delta_{t,t'} + \alpha^2\sum_{t''=0}^{t} R_x[t,t'']R_x[t'',t']Q''\left(C_x[t,t'']\right)$$

$$- \alpha\mu\left(C_x[t,t]-1\right)R_x[t,t']. \tag{27}$$

with initial conditions: $C_x[0,0] = 1$, and $R_x[t+1,t] = 1$. Apart from the $\mu$-dependent term, these equations are a particular case of the ones that appear in [36, 37] and we point to these previous references for details.

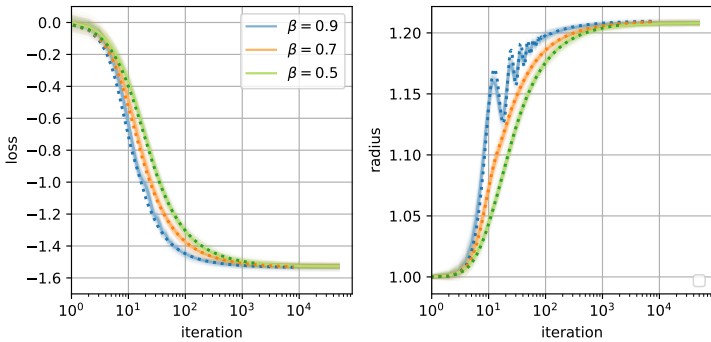

Figure 2: **DMFT for HB.** Simulations of HB momentum in the mixed $p$-spin model with $p = 3$, $\Delta_3 = 2/p$, $\Delta_2 = 1$, ridge parameter $\mu = 10$ and input dimension $N = 1024$. The parameters are $\alpha = 0.01$ for all the simulations and $\beta \in \{0.5, 0.7, 0.9\}$. We use solid lines to represent the result from the simulation, the dotted lines for the DMFT of HB.

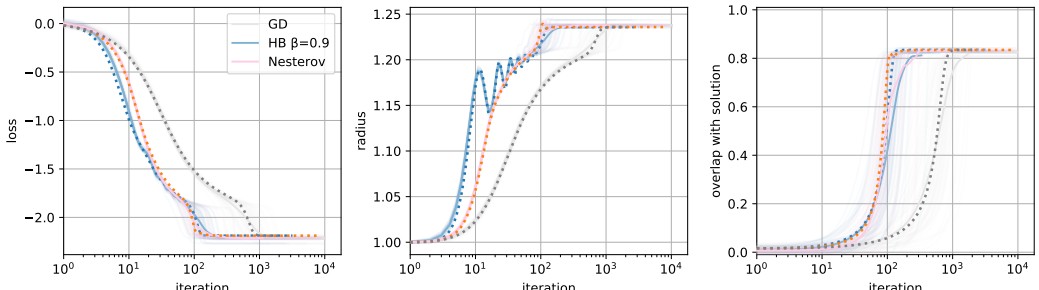

Figure 3: **DMFT in the spiked matrix-tensor model.** Performance of heavy ball and Nesterov in the spiked matrix-tensor model with $p = 3$, $1/\Delta_2 = 2.7$, $\Delta_3 = 1.0$, and $\mu = 10$. The parameters in the simulations are: $\alpha = 0.01$ and $\beta = 0.9$ for HB. The different solid lines correspond to simulations with input dimension $N = 8192$, while the dotted lines are obtained from the DMFT that, by definition, is in the infinite dimension limit. In the spiked version of the model the finite size effects are stronger and larger simulation sizes are needed.

## 4    Derivation of DMFT for Nesterov acceleration

The approach for the DMFT proposed in this section is based on the dynamical cavity method [43]. Consider the problem having dimension $N + 1$ and denote the additional entry of the vectors $\boldsymbol{x}$ and $\boldsymbol{y}$ with the subscript $0$, $x_0$ and $y_0$. The idea behind cavity method is to evaluate how this additional dimension changes the dynamics of all degrees of freedom. If the dimension is sufficiently large the dynamics is only slightly modified by the additional dimension, and the effect of the additional degree of freedom can be tracked in perturbation theory.

The framework described in this section might be extended to more other momentum-based algorithms (such as PID [11] and quasi-hyperbolic momentum [12]) with some minor adaptations. The steps to follow [43] can be summarised in:

- Writing the equation of motion isolating the contributions of an additional degree of freedom, leading to Eqs. (28-30;
- Treating the effect of the terms containing the new degree of freedom in perturbation theory, Eqs. (32-34);
- Identifying the order dynamical order parameters, namely dynamical correlation and response functions, Eqs. (37,38).

Consider the Nesterov update algorithm and isolate the effect of the additional degree of freedom

$$x_i[t+1] = y_i[t] + \alpha \sum_{j\neq 0} J_{ij} y_j[t] + \alpha \sum_{(i,i_2,\dots,i_p)} J_{i,i_2,\dots,i_p} y_{i_2}[t] \dots y_{i_p}[t] - \alpha\mu \left( \sum_{j\neq 0} \frac{y_j^2[t]}{N} - 1 \right) y_i[t] \tag{28}$$

$$+ \alpha \sum_{(i,0,i_3,\dots,i_p)} J_{i,0,i_3,\dots,i_p} y_0[t] y_{i_3}[t] \dots y_{i_p}[t] + \alpha J_{i0} y_0[t] + \frac{\mu}{N} y_0^2[t] y_i[t], \tag{29}$$

$$y_i[t+1] = x_i[t+1] + \frac{t}{t+3} \left( x_i[t+1] - x_i[t] \right). \tag{30}$$

We identify the term in line (29) as a perturbation, denoted by $H_i[t]$. We will assume that the perturbation is sufficiently small and the effective dynamics is well approximated by a first order expansion around the original updates, so-called *linear response regime*. Therefore, the perturbed entries can be written as

$$x_i[t] \approx x_i^0 + \alpha \sum_{t''=0}^{t} \frac{\delta x_i[t]}{\delta H_i[t'']} H_i[t''], \qquad y_i[t] \approx y_i^0 + \alpha \sum_{t''=0}^{t} \frac{\delta y_i[t]}{\delta H_i[t'']} H_i[t'']. \tag{31}$$

The dynamics of the 0th degree of freedom to the leading order in the perturbation is

$$x_0[t+1] = y_0[t] - \alpha\mu\Big(\frac{1}{N}\sum_j y_j^2[t] - 1\Big)y_0[t] + \Xi[t] + \alpha^2\sum_j J_{0j}\sum_{t''=0}^t \frac{\delta y_j[t]}{\delta H_j[t'']}H_j[t''] \quad (32)$$

$$+ \alpha^2 \sum_{(0,i_2,\dots,i_p)} J_{0,i_2,\dots,i_p}\Big(\sum_{t''=0}^t \frac{\delta y_{i_2}[t]}{\delta H_{i_2}[t'']}H_{i_2}[t'']y_{i_3}[t]\dots y_{i_p}[t] + \text{perm.}\Big) + \mathcal{O}\Big(\frac{1}{N}\Big), \quad (33)$$

$$y_i[t+1] = x_i[t+1] + \frac{t}{t+3}\left(x_i[t+1] - x_i[t]\right), \quad (34)$$

with $\Xi = \alpha\sum_j J_{0j}y_j[t] + \alpha\sum_{(0,i_2,\dots,i_p)} J_{0,i_2,\dots,i_p}y_{i_2}[t]\dots y_{i_p}[t]$ a Gaussian noise with moments:

$$\mathbb{E}[\Xi[t]] = 0,$$

$$\mathbb{E}[\Xi[t]\Xi[t']] = \frac{1}{\Delta_2}C_y[t,t'] + \frac{1}{\Delta_p}C_y^{p-1}[t,t'] = Q'\left(C_y[t,t']\right) \doteq \mathbb{K}[t,t'].$$

The terms in Eqs. (32,33) can be simplified. Consider the last term in Eq. (32): after substituting the $H_i$, $J_{0j}J_{0j}$ and $J_{0j}J_{(j,0,\dots,i_p)}$ can be approximated by their expected values with a difference that is subleading in $1/N$

$$\alpha^2\sum_j J_{0j}\sum_{t''=0}^t \frac{\delta y_j[t]}{\delta H_j[t'']}J_{0j}y_0[t''] \approx \frac{\alpha^2}{\Delta_2 N}\sum_{t''=0}^t \frac{\delta y_j[t]}{\delta H_j[t'']}y_0[t''] = \frac{\alpha^2}{\Delta_2}\sum_{t''=0}^t R_y[t,t'']y_0[t''], \quad (35)$$

where the last equality follows from the definition of response function in $y$.
The same approximation is applied to Eq. (33), taking carefully into account the permutations, obtaining

$$\frac{\alpha^2(p-1)}{\Delta_p}\sum_{t''=0}^t R_y[t,t'']\left(C_y[t,t'']\right)^{p-2}y_0[t'']. \quad (36)$$

Finally, collecting all terms, the effective dynamics of the additional dimension is given by

$$x_0[t+1] = y_0[t] + \alpha\Xi[t] - \alpha\mu\left(C_y[t,t]-1\right)y_0[t] + \alpha^2\sum_{t''=0}^t R_y[t,t'']Q''\left(C_y[t,t'']\right)y_0[t'']; \quad (37)$$

$$y_0[t+1] = x_0[t+1] + \frac{t}{t+3}\left(x_0[t+1] - x_0[t]\right). \quad (38)$$

In order to derive the updates of the order parameters, we need the expected values of $\langle\Xi[t]x_0[t']\rangle$ and $\langle\Xi[t]y_0[t']\rangle$ with respect to the stochastic process. These are obtained using Girsanov theorem

$$\langle\Xi[t]x_0[t']\rangle = \alpha\sum_{t''} R_x[t',t'']Q'\left(C_y[t,t'']\right), \qquad \langle\Xi[t]y_0[t']\rangle = \alpha\sum_{t''} R_y[t',t'']Q'\left(C_y[t,t'']\right).$$

The final step consists in substituting the Eqs. (37,38) into the equations of the order parameters Eqs. (14-19). Then we identify the order parameters in the equations and use the results of Girsanov theorem to obtain the dynamical equations reported in section 3.

## 5    Algorithmic threshold

Finally we investigate the performance of accelerated methods in recovering a signal in a complex non-convex landscape. The dynamics of the gradient descent has been studied in the spiked matrix-tensor model in [33]. Using DMFT it was possible to compute the phase diagram for signal recovery in terms of the noise levels $\Delta_2$ and $\Delta_p$. This phase diagram was later confirmed theoretically [34].

Given the DMFT equations derived in the previous sections we can apply the analysis used in [33] to accelerated gradient methods. Given order of the tensor $p$ and $\Delta_p$, increasing $\Delta_2$ the problem becomes harder and moves from the easy phase - where the signal can be partially recovered - to an algorithmically impossible phase - where the algorithm remains stuck at vanishingly small overlap

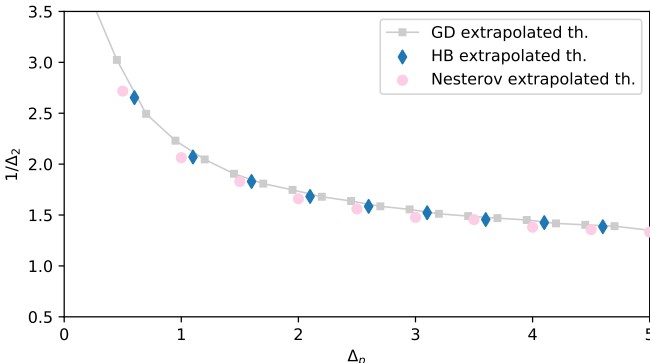

Figure 4: **Phase diagram of the spiked matrix-tensor model.** The horizontal and vertical axis represent the parameters of the model $\Delta_p$ and $1/\Delta_2$. We identify two regions in the diagram: where Nesterov, heavy ball and gradient descent algorithms lead to the hidden solution (upper region), and where they fail (lower region). The grey square connected by a solid line represents the threshold of gradient descent estimated numerically as detailed in the text. We use points to indicate the threshold extrapolated from the DMFT: pink circles for Nesterov acceleration and blue diamonds for heavy ball momentum with $\beta = 0.9$ and $\alpha = 0.01$.

with the signal. The goal of the analysis is to characterize the algorithmic threshold that separates the two phases. Using the DMFT we estimate the *relaxation time* – the time the accelerated methods need to find the signal. Since this time diverges approaching the algorithmic threshold, the fit of the divergence point gives an estimation of the threshold.

More precisely, for each value of $\Delta_p$ as the noise to signal ratio ($\Delta_2$) increases the simulation time required to arrive close to the signal[1] increases like a power law $\sim a\,|\Delta_2 - \Delta_2^{al\cdot}(\Delta_p)|^{-\theta}$. The algorithmic threshold $\Delta_2^{al\cdot}(\Delta_p)$ is obtained by fitting the parameters of the power law $(a, \theta, \Delta_2^{al\cdot})$. In the SM we show an example of the extrapolation of a single point where many initial conditions $m_x(0)$ are considered in order to correctly characterize the limits $N \to \infty$ and $m_x(0) \to 0^+$. Finally the fits obtained for the three algorithms and for several $\Delta_p$ are shown in the phase diagram of Fig. 4 for $p = 3$. We observe that all the algorithms give very close thresholds. DMFT allows to obtain a good estimation of the threshold, free from finite size effects and stochastic fluctuations that are present in the direct estimation from the simulations.

## Conclusions and broader impact

In this work we analysed momentum-accelerated methods in two paradigmatic high-dimensional non-convex problems: the mixed $p$-spin model and the spiked matrix-tensor model. Our analysis is based on dynamical mean field theory and provides a set of equations that characterize the average evolution of the dynamics. We have focused on Polyak's heavy ball and Nesterov acceleration, but the same techniques may be applied to more recent methods such as quasi-hyperbolic momentum [12] and proportional integral-derivative control algorithm [11].

Momentum-based methods are techniques commonly used in practice but poorly understood at the theoretical level. This work analysed the dynamics of momentum-based algorithms in a very controlled setting of a high-dimensional non-convex inference problem which allowed us to establish that accelerated methods have a recovery threshold which is – within the limits of numerical integration – the same of vanilla gradient descent.

Our analysis can be easily extended to 1-layer neural networks – combining our technical results with the techniques of [46] – and to simple inference problem seen from the learning point of view, such as the phase retrieval problem [53]. The same questions can also be analysed in the context of recurrent networks [48, 49] where DMFT approaches have already been applied to gradient-based methods.

---

[1]Since the best possible overlap for maximum a posteriori estimator $m^{\text{MAP}}$ can be computed explicitly, "close" means the time that the algorithms takes to arrive at $0.9 m^{\text{MAP}}$

Our study is theoretical in nature and we do not foresee any societal impact.

## Acknowledgments

The authors thank Andrew Saxe for precious discussions. This work was supported by the Wellcome Trust and Royal Society (grant number 216386/Z/19/Z), and by "Investissements d'Avenir" LabEx-PALM (ANR-10-LABX-0039-PALM).

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
