# Supplemental Material

## A  Spiked matrix-tensor model

In this section we discuss how the DMFT equations of the spiked matrix-tensor model differ from the mixed $p$-spin model. The main difference is that the hidden solution deforms locally the loss function

$$
\mathcal{H} = -\frac{1}{\Delta_p}\sqrt{\frac{(p-1)!}{N^{p-1}}}\sum_{i_1,\ldots,i_p=1}^{N} T_{i_1,\ldots,i_p}x_{i_1}\ldots x_{i_p} - \frac{1}{\Delta_2}\frac{1}{\sqrt{N}}\sum_{i,j=1}^{N}Y_{ij}x_ix_j +
$$

$$
-\frac{1}{p\Delta_p}\left(\frac{1}{N}\sum_j x_j x_j^*\right)^p - \frac{1}{2\Delta_2}\left(\frac{1}{N}\sum_j x_j x_j^*\right)^2 + \frac{\mu}{4N}\left(\sum_{i=1}^{N}x_i^2 - N\right)^2. \tag{39}
$$

As it clearly appears from the equation of the loss, the overlap of the hidden solution with the estimator plays an important role. This leads to two additional order parameters $m_x[t] = \frac{1}{N}\sum_j x_j[t]x_j^*$ and $m_y[t] = \frac{1}{N}\sum_j y_j[t]x_j^*$ (or $m_v(t) = \frac{1}{N}\sum_j v_j(t)x_j^*$ for massive gradient flow).

Since the stochastic part of the loss is unchanged, the derivation follows same steps shown in section 4 of the main text. They lead to modified dynamical equations where overlap with the hidden solution is present, for instance in Nesterov they are

$$
x_0[t+1] = y_0[t] + \alpha\Xi[t] - \alpha\mu\left(C_y[t,t] - 1\right)y_0[t] +
$$

$$
+\alpha^2\sum_{t''=0}^{t}R_y[t,t'']Q''\left(C_y[t,t'']\right)y_0[t''] + Q'(m_y[t])\boldsymbol{x}^*, \tag{40}
$$

$$
y_0[t+1] = x_0[t+1] + \frac{t}{t+3}\left(x_0[t+1] - x_0[t]\right). \tag{41}
$$

Finally, substituting the effective dynamics into the definition of the order parameters we obtain:

- **for Nesterov acceleration**

$$
C_x[t+1,t'] = C_{xy}[t,t'] + \alpha^2\sum_{t''=0}^{t'}R_x[t',t'']Q'\left(C_y[t,t'']\right) + \alpha^2\sum_{t''=0}^{t}R_y[t,t'']Q''\left(C_y[t,t'']\right)C_{xy}[t',t''] +
$$

$$
-\alpha\mu\left(C_y[t,t]-1\right)C_y[t,t'] - Q'(m_y[t])m_x[t'],
$$

$$
C_{xy}[t+1,t'] = C_y[t,t'] + \alpha^2\sum_{t''=0}^{t'}R_y[t',t'']Q'\left(C_y[t,t'']\right) + \alpha^2\sum_{t''=0}^{t}R_y[t,t'']Q''\left(C_y[t,t'']\right)C_y[t',t''] +
$$

$$
-\alpha\mu\left(C_y[t,t]-1\right)C_{xy}[t,t'] - Q'(m_y[t])m_y[t'],
$$

$$
C_{xy}[t',t+1] = C_x[t+1,t'] + \frac{t}{t+3}\left(C_x[t+1,t'] - C_x[t,t']\right),
$$

$$
C_y[t',t+1] = C_{xy}[t+1,t'] + \frac{t}{t+3}\left(C_{xy}[t+1,t'] - C_{xy}[t,t']\right),
$$

$$
R_x[t+1,t'] = R_y[t,t'] + \delta_{t,t'} + \alpha^2\sum_{t''=t'}^{t}R_y[t,t'']R_y[t'',t']Q''\left(C_y[t,t'']\right) - \alpha\mu\left(C_y[t,t]-1\right)R_y[t,t'],
$$

$$R_y[t', t+1] = R_x[t+1, t'] + \frac{t}{t+3}\left(R_x[t+1, t'] - R_x[t, t']\right),$$

$$m_x[t+1] = m_y[t] - \alpha\mu\left(C_y[t,t] - 1\right)m_y[t] + \alpha^2\sum_{t''=0}^{t}R_y[t,t'']Q''\left(C_y[t,t'']\right)m_y[t''] + Q'\left(m_y[t]\right),$$

$$m_y[t+1] = m_x[t+1] + \frac{t}{t+3}\left(m_x[t+1] - m_x[t]\right),$$

with initial conditions $C_x[0,0] = 1$, $C_y[0,0] = 1$, $C_{xy}[0,0] = 1$, $R_x[t+1,t] = 1$, $R_y[t+1,t] = \frac{2t+3}{t+3}$, $m_x[0] = 0^+$, $m_y[0] = 0^+$;

- **for heavy ball momentum.**

$$C_y[t+1, t'] = \beta C_y[t,t'] + \mu\left(C_x[t,t] - 1\right)C_{xy}[t,t'] + \alpha\sum_{t''=0}^{t'}R_y[t',t'']Q'\left(C_x[t,t'']\right)$$

$$+ \alpha\sum_{t''=0}^{t}R_x[t,t'']Q''\left(C_x[t,t'']\right)C_{xy}[t'',t'] - Q'(m_x[t])m_y[t'];$$

$$C_{xy}[t', t+1] = \beta C_{xy}[t',t] + \mu\left(C_x[t,t] - 1\right)C_x[t,t'] + \alpha\sum_{t''=0}^{t'}R_x[t',t'']Q'\left(C_x[t,t'']\right)$$

$$+ \alpha\sum_{t''=0}^{t}R_x[t,t'']Q''\left(C_x[t,t'']\right)C_x[t',t''] - Q'(m_x[t])m_x[t'];$$

$$C_{xy}[t+1, t'] = C_{xy}[t,t'] - \alpha C_y[t+1,t'];$$

$$C_x[t+1, t'] = C_x[t,t'] - \alpha C_{xy}[t', t+1];$$

$$R_y[t+1, t'] = \beta R_y[t,t'] + \frac{1}{\alpha}\delta_{t,t'} + \mu\left(C_x[t,t] - 1\right)R_x[t,t']$$

$$+ \alpha\sum_{t''=0}^{t}R_x[t,t'']R_x[t'',t']Q''\left(C_x[t,t'']\right);$$

$$R_x[t+1, t'] = R_x[t,t'] - \alpha R_y[t+1,t']$$

$$m_y[t+1] = \beta m_y[t] - \mu\left(C_x[t,t] - 1\right)m_x[t] +$$

$$+ \sum_{t''=0}^{t}R_x[t,t'']Q''\left(C_x[t,t'']\right)m_x[t''] - Q'\left(m_x[t]\right),$$

$$m_x[t+1] = m_x[t] - \alpha m_y[t+1].$$

with initial conditions: $C_x[0,0] = 1$, $C_y[0,0] = 0$, $C_{xy}[0,0] = 0$, $R_y[t+1,t] = 1/\alpha$, $R_x[t+1,t] = -1$, $m_y[0] = O^+$, $m_x[0] = O^+$.

- **for massive gradient flow** (see Sec. C)

$$\partial_t C_x(t, t') = C_{xv}(t', t),$$

$$m\partial_t C_v(t, t') = -C_v(t,t') + \int_0^{t}dt''R_{x|v}(t,t'')Q''[C_x(t,t'')]C_{xv}(t'',t') +$$

$$+ \int_0^{t'}Q'[C_x(t,t'')]R_v(t',t'') - \mu C_x(t.t')\left(C_x(t,t) - 1\right) + Q'[m_x(t')]m_x(t),$$

$$\partial_t C_{xv}(t, t') = C_v(t,t'),$$

$$m\partial_{t'}C_{xv}(t, t') = -C_{xv}(t,t') + \int_0^{t'}dt''R_{x|v}(t',t'')Q''[C_x(t',t'')]C_x(t,t'') +$$

$$+ \int_0^{t}Q'[C_x(t',t'')]R_{x|v}(t,t'') - \mu C_{xv}(t,t')\left(C_x(t,t) - 1\right) + Q'[m_x(t)]m_v(t'),$$

$$m\partial_t R_v(t, t') = \delta(t - t') - R_v(t, t') + \int_{t'}^{t} dt'' Q''[C(t, t'')] R_{x|v}(t, t'') R_{x|v}(t'', t') +$$
$$- \mu R_{x|v}(t, t') \left(C_x(t, t) - 1\right),$$
$$\partial_t R_{x|v}(t, t') = R_v(t, t'),$$
$$\partial_t m_x(t) = m_v(t),$$
$$m\partial_t m_v(t) = -m_v(t) + \int_0^t dt'' R_{x|v}(t, t'') Q''[C_x(t, t'')] m_x(t'') + Q'[m_x(t)] +$$
$$- \mu m_x(t) \left(C_x(t, t) - 1\right),$$

with initial conditions are : $C_x(0,0) = 1$; $C_v(0,0) = 0$; $C_{xv}(0,0) = 0$; $R_v(t^+, t) = 1/m$; $R_{x|v}(t, t) = 0$; $m_x(0) = 0^+$. $m_y(0) = 0^+$.

Finally the equation to compute the loss in time is

$$\mathcal{L}[t] = -\frac{\alpha}{\Delta_p C_x[t,t]^{\frac{p}{2}}} \sum_{t''=0}^{t} R_x[t,t'] C_x[t,t']^{p-1} - \frac{\alpha}{\Delta_2 C_x[t,t]} \sum_{t''=0}^{t} R_x[t,t'] C_x[t,t'] - Q\left(m_x[t]\right).$$
$$(42)$$

## B  Extracting the recovery threshold

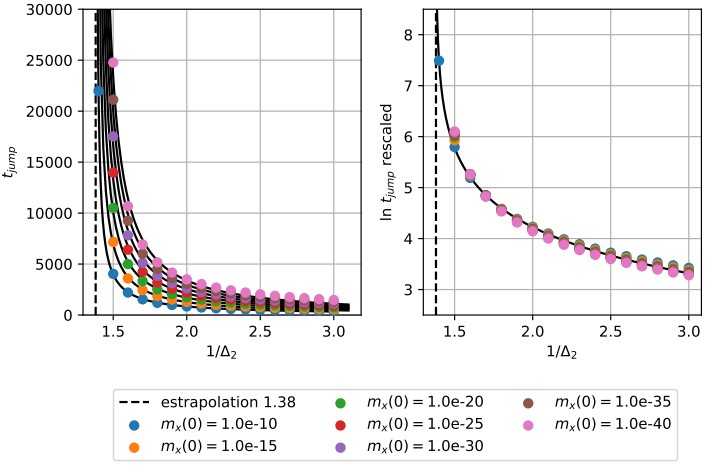

Figure 5: **Algorithmic threshold extrapolated using Nesterov acceleration.** The dots are the divergence time obtained for the different values of $\Delta_2$ with $\Delta_3 = 4.0$ fixed. On the right panel we show that these lines collapse to a single line once rescaled by a factor $a^{\ln m_x[0]}$ with $a \approx 1.089$.

The extrapolation procedure for $\Delta_3 = 4.0$ is shown in Fig. 5. The threshold obtained by fitting with a power law and observing the divergent $\Delta_2$. On the left panel we plot the number of iteration before the algorithm jumps to the solution $t_{jump}$ as a function of the signal to noise ratio $1/\Delta_2$. The figure also show remarkable effects of the initial conditions for $m_x$ and $m_y$. These effects where already described and understood in [33].

## C Correspondence with continuous HB equations

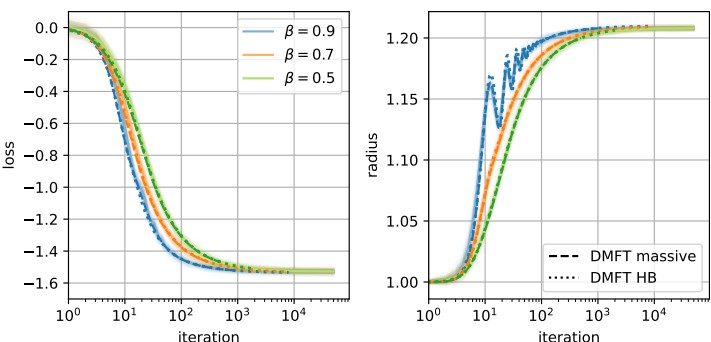

Figure 6: **Comparison of HB and massive with mapping.** The figure reproduce the same setting of Fig. 2 with the additional dashed line for the DMFT of massive gradient flow using the mapping.

An alternative way to analyze the HB dynamics is by using the results of [51] to map it to the massive momentum described by the flow equation

$$m\ddot{x}_i(t) + \dot{x}_i(t) = -\frac{\delta\mathcal{L}[\boldsymbol{x}(t)]}{\delta x_i(t)}. \tag{43}$$

The natural discretization of this equation is

$$\frac{m}{h^2}\left(x[k+1] - 2x[k] - x[k-1]\right) + \frac{1}{h}\left(x[k+1] - x[k]\right) = -\nabla\mathcal{L}(x[k]) \tag{44}$$

being $h$ the time discretization step (analogous to the learning rate in gradient descent). Using the mapping of [51] we can identify

$$m = \frac{\beta\alpha}{(1-\beta)^2} \tag{45}$$

$$h = \frac{\alpha}{1-\beta} \tag{46}$$

Observe that in order to be consistent with a continuous dynamics we need the following scaling $\beta = \mathcal{O}(1)$, $\alpha = \mathcal{O}[(1-\beta)^2]$. We empirically observe in the simulations a good agreement between massive and HB even for $\beta = 0.999$ and $\alpha = 0.01$. In the following, when discussing the comparison between simulation and DMFT, we mean that we run HB algorithm and superimpose on its massive momentum description.

The massive momentum dynamics was also considered in [52] without the damping term $(\dot{x}_i(t))$ and for the model with a hard spherical constraint $\sum_i x_i[t]^2 = N$. While the DMFT derived in [52] completely describe the aforementioned particular case, the way in which it is written uses the fact that without damping the dynamics is conservative and the spherical constraint can be enforced using that. In our case we are not in this regime and therefore we resort to a different computation, that will lead us to quite different equations. Indeed if one wants to transform massive momentum in a practical algorithm one needs to transform the second order ODEs into first order by defining velocity variables $v_i(t) = \dot{x}_i(t)$. Then the discrete version of Eq. (43) is

$$x_i[t+1] = x_i[t] + hv_i[t]$$
$$v_i[t+1] = v_i[t] - \frac{h}{m}\,v_i[t] - \frac{h}{m}\,\frac{\delta\mathcal{H}}{\delta x_i[t]} \tag{47}$$

Analysing these equations through DMFT one gets a set of flow equations for the following dynamical order parameters $C_x[t,t'] = \sum_i x_i[t]x_i[t']/N$, $C_v[t,t'] = \sum_i v_i[t]v_i[t']/N$, $C_{xv}[t,t'] = \sum_i x_i[t]v_i[t']/N$, $R_v[t,t'] = \frac{1}{N}\sum_i \frac{\delta v_i[t]}{\delta H_i[t']}$, and $R_{x|v}[t,t'] = \frac{1}{N}\sum_i \frac{\delta x_i[t]}{\delta H_i[t']}$; where $\boldsymbol{H}$ is an instantaneous perturbation acting on the velocity. The result of the computation gives:

$$C_x[t+1,t'] = C_x[t,t'] + h\,C_{xv}(t',t)\,; \tag{48}$$

$$C_v[t+1,t'] = C_v[t,t'] - \frac{h}{m} C_v(t,t') - \mu \frac{h}{m} C_{xv}(t,t') \left(C_x(t,t) - 1\right)$$

$$+ \frac{h^2}{m} \sum_{t''=0}^{t} R_{x|v}[t,t'']Q'' \left(C_x[t,t'']\right) C_{xv}[t'',t'] + \frac{h^2}{m} \sum_{t''=0}^{t'} Q' \left(C_x[t,t'']\right) R_v[t',t'']; \tag{49}$$

$$C_{xv}[t+1,t'] = C_{xv}[t,t'] + h \, C_v[t,t'] \; ; \tag{50}$$

$$C_{xv}[t,t'+1] = C_{xv}[t,t'] - \frac{h}{m} C_{xv}[t,t'] - \mu \frac{h}{m} C_x[t,t'] \left(C_x[t',t'] - 1\right)$$

$$+ \frac{h^2}{m} \sum_{t''=0}^{t'} R_{x|v}[t',t'']Q'' \left(C_x[t',t'']\right) C_x[t,t''] + \frac{h^2}{m} \sum_{t''=0}^{t} Q' \left(C_x[t',t'']\right) R_{x|v}[t,t''] \tag{51}$$

$$R_v[t+1,t'] = R_v[t,t'] + \frac{h}{m} \delta_{t,t'} - \mu \frac{h}{m} R_{x|v}[t,t'] \left(C_x(t,t) - 1\right) - \frac{h}{m} R_v[t,t']$$

$$+ \frac{h^2}{m} \sum_{t''=0}^{t} Q'' \left(C[t,t'']\right) R_{x|v}[t,t'']R_{x|v}[t'',t'] \tag{52}$$

$$R_{x|v}[t+1,t'] = R_{x|v}[t,t'] + h \, R_v[t,t']. \tag{53}$$

and initial conditions : $C_x[0,0] = 1, C_v[0,0] = 0, C_{xv}[0,0] = 0, R_v[t+1,t] = 1/m, R_{x|v}[t+1,t] = 0$.

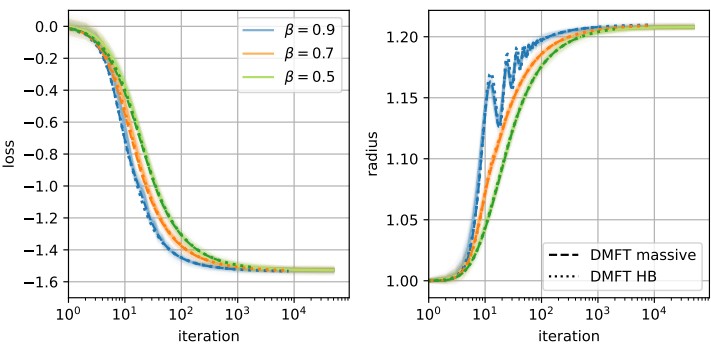

Figure 7: **Comparison of HB and with DMFT.** Simulations of HB momentum in the mixed $p$-spin model with $p = 3$, $\Delta_3 = 2/p$, $\Delta_2 = 1$, ridge parameter $\mu = 10$ and input dimension $N = 1024$. The parameters are $\alpha = 0.01$ for all the simulations and $\beta \in \{0.5, 0.7, 0.9, 0.99, 0.999\}$. We use solid line to represent the result from the simulation, the dotted line for the DMFT of massive gradient flow with the mapping. We empirically observe that in this problem the value of $\beta$ that gives the best speed up is $\beta = 0.9$. In order the integrate the DMFT of massive gradient we matched the mass as described in Eq. (45) and consider time steps $h \in \{0.005, 0.005, 0.0125, 0.25, 0.25\}$.

**Derivation of the DMFT equations for massive gradient flow**  In this section we derive the dynamical mean field theory (DMFT) equations of massive gradient flow in the mixed $p$-spin.

We use the generating functional approach described in [54, 55] to obtain the effective dynamical equations. First we rewrite de massive dynamics Eq. (43) as two ODEs

$$m\dot{\boldsymbol{v}}(t) = -\boldsymbol{v}(t) - \nabla \mathcal{H}[\boldsymbol{x}(t)], \tag{54}$$

$$\dot{\boldsymbol{x}}(t) = \boldsymbol{v}(t). \tag{55}$$

We use the following simple identity that takes the name of *generating functional*

$$1 = \mathcal{Z} = \int \mathcal{D}[\boldsymbol{x}, \boldsymbol{v}] \, \delta \left(m\dot{\boldsymbol{v}}(t) + \boldsymbol{v}(t) + \nabla \mathcal{H}[\boldsymbol{x}(t)]\right) \, \delta \left(\dot{\boldsymbol{x}}(t) - \boldsymbol{v}(t)\right) \tag{56}$$

$$= \int \mathcal{D}[\boldsymbol{x}, \tilde{\boldsymbol{x}}, \boldsymbol{v}, \tilde{\boldsymbol{v}}] \prod_{i=1}^{N} \exp \left\{ i \int \tilde{v}_i(t) \left[m\dot{v}_i(t) + v_i(t) + \nabla_i \mathcal{H}[\boldsymbol{x}(t)]\right] dt \right\} \exp \left\{ i \int \tilde{x}_i(t) \left[\dot{x}_i(t) - v_i(t)\right] dt \right\} \tag{57}$$

where in the first line we integrate over all possible trajectories of $\boldsymbol{v}$ and $\boldsymbol{x}$, and we impose them to match the massive gradient flow equations using Dirac's deltas. In the second line we use the Fourier representation of the delta and we absorb the normalization constants in the term $\mathcal{D}[\boldsymbol{x}, \tilde{\boldsymbol{x}}, \boldsymbol{v}, \tilde{\boldsymbol{v}}]$. We can now average over the stochasticity of the problem, let us indicate with an overline the average.

$$1 = \overline{\mathcal{Z}} = \int \mathcal{D}[\boldsymbol{x}, \tilde{\boldsymbol{x}}, \boldsymbol{v}, \tilde{\boldsymbol{v}}] \prod_{i=1}^{N} \exp\left\{ i \int \tilde{v}_i(t) \left[ m\dot{v}_i(t) + v_i(t) \right] dt \right\} \exp\left\{ i \int \tilde{x}_i(t) \left[ \dot{x}_i(t) - v_i(t) \right] dt \right\} \times$$

$$(58)$$

$$\times \overline{ \prod_{i=1}^{N} \exp\left\{ i \int \tilde{v}_i(t) \left[ \sqrt{\frac{(p-1)!}{N^{p-1}}} \sum_{(i,i_2,\dots,i_p)} \xi^{(p)}_{i,i_2,\dots,i_p} x_{i_2}(t) \dots x_{i_p}(t) + \frac{1}{\sqrt{N}} \sum_j \xi^{(2)}_{i,j} x_j(t) \right] dt \right\} } \times$$

$$(59)$$

$$\times \prod_{i=1}^{N} \exp\left\{ i \int \tilde{v}_i(t) \mu \left( \frac{1}{N} \sum_j x_j^2(t) - 1 \right) x_i(t) dt \right\}$$

$$(60)$$

We can proceed integrating the second line over the noise. Importantly we must group all the element that multiply a given $\xi^{(p)}$. Considering only second line and neglecting constant multiplicative factors we obtain

$$\exp\left\{ -\frac{N}{2p\Delta_p} \int \left[ p \left( \sum_i \frac{\tilde{v}_i(t)\tilde{v}_i(t')}{N} \right) \left( \sum_i \frac{x_i(t)x_i(t')}{N} \right)^{p-1} + \right.\right.$$

$$\left.\left. + p(p-1) \left( \sum_i \frac{\tilde{v}_i(t)x_i(t')}{N} \right) \left( \sum_i \frac{x_i(t)\tilde{v}_i(t')}{N} \right) \left( \sum_i \frac{x_i(t)x_i(t')}{N} \right)^{p-2} \right] dt' dt \right\} \times$$

$$\times \exp\left\{ -\frac{N}{2\Delta_2} \int \left[ \left( \sum_i \frac{\tilde{v}_i(t)\tilde{v}_i(t')}{N} \right) \left( \sum_i \frac{x_i(t)x_i(t')}{N} \right) + \left( \sum_i \frac{\tilde{v}_i(t)x_i(t')}{N} \right) \left( \sum_i \frac{x_i(t)\tilde{v}_i(t')}{N} \right) \right] dt' dt \right\}$$

We define $Q(x) = x^p/(p\Delta_p) + x^2/(2\Delta_2)$ and the order parameters

$$C_x[t, t'] = \sum_i x_i[t] x_i[t']/N,$$

$$(61)$$

$$C_v[t, t'] = \sum_i v_i[t] v_i[t']/N,$$

$$(62)$$

$$C_{xv}[t, t'] = \sum_i x_i[t] v_i[t]/N,$$

$$(63)$$

$$R_v[t, t'] = \frac{1}{N} \sum_i \frac{\delta v_i[t]}{\delta H_i[t']},$$

$$(64)$$

$$R_{x|v}[t, t'] = \frac{1}{N} \sum_i \frac{\delta x_i[t]}{\delta H_i[t']};$$

$$(65)$$

and enforce some of them using Dirac's deltas

$$\int \mathcal{D}[C_x, C_{\tilde{v}}, C_{x\tilde{v}}, C_{\tilde{v}x}] \delta \left( NC_{\tilde{v}}(t, t') - \sum_j i\, \tilde{v}_j(t)\, i\, \tilde{v}_j(t') \right) \delta \left( NC_x(t, t') - \sum_j x_j(t)\, x_j(t') \right) \times$$

$$\times \delta \left( NC_{x\tilde{v}}(t, t') - \sum_j x_j(t)\, i\, \tilde{v}_j(t') \right) \delta \left( NC_{\tilde{v}x}(t, t') - \sum_j i\, \tilde{v}_j(t)\, x_j(t') \right) \times$$

$$\times \exp\left\{ -\frac{N}{2\Delta_p} \int \left[ C_{\tilde{v}}(t, t') Q'\left[ C_x(t, t') \right] + C_{\tilde{v}x}(t, t') C_{x\tilde{v}}(t, t') Q''\left[ C_x(t, t') \right] \right] dt' dt \right\}.$$

Using again the Fourier representation of the deltas and considering $N$ large, the auxiliary variables introduced with the transform concentrate to their saddle point according to Laplace approximation. Furthermore it is easy to show [54] that $C_{x\tilde{v}}(t,t') = C_{\tilde{v}x}(t',t)$ and $C_{x\tilde{v}}(t,t') = R_{x|v}(t,t')$, with $R_{x|v}$. Under this considerations, we rewrite the average generating functional

$$1 = \overline{\mathcal{Z}} = \int \mathcal{D}[\boldsymbol{x},\tilde{\boldsymbol{x}},\boldsymbol{v},\tilde{\boldsymbol{v}}] \prod_{i=1}^{N} \exp\left\{ i \int \tilde{v}_i(t) \left[ m\dot{v}_i(t) + v_i(t) + \mu \left( \frac{\sum_j x_j^2(t)}{N} - 1 \right) x_i(t) \right] dt \right\} \times$$
(66)

$$\times \prod_{i=1}^{N} \exp\left\{ i \int \tilde{x}_i(t) \left[ \dot{x}_i(t) - v_i(t) \right] dt \right\} \times$$
(67)

$$\times \exp\left\{ -\int \sum_j \left[ \frac{1}{2} Q'[C_x(t,t')]i\,\tilde{v}_j(t)\,i\,\tilde{v}_j(t') - Q''[C_x(t,t')]R_{x|v}(t,t')\,i\,\tilde{v}_j(t')\,x_j(t) \right] dt' dt \right\}.$$
(68)

Finally we can take a Hubbard-Stratonovich transform on the first term of the second line and identify a stochastic process $\Xi(t)$ with zero mean at all times and covariance $\mathbb{E}[\Xi(t)\Xi(t')] = Q'[C_x(t,t')]$. The resulting generating functional now represent the dynamics in $\boldsymbol{x}$ and $\boldsymbol{v}$ where the cross-element interactions are replaced by the stochastic process. The resulting effective equations are

$$m\dot{\boldsymbol{v}}(t) = -\boldsymbol{v}(t) + \int_0^t dt'' R_{x|v}(t,t'')Q'[C_x(t,t'')]\boldsymbol{x}(t'') + \boldsymbol{\Xi}(t) - \mu\left[C_x(t,t) - 1\right]\boldsymbol{x}(t),$$
(69)

$$\dot{\boldsymbol{x}}(t) = \boldsymbol{v}(t).$$
(70)

Finally we introduce the order parameters Eqs. (61-65) and compute their equation explicitly by substituting the equations of the effective dynamics. In order to do that, we use Girsanov theorem and evaluate the following expected values

$$\langle v(t)\Xi(t') \rangle = \int_0^{t'} dt'' R_v(t,t'')Q'[C_x(t,t'')],$$
(71)

$$\langle x(t)\Xi(t') \rangle = \int_0^{t'} dt'' R_{x|v}(t,t'')Q'[C_x(t,t'')].$$
(72)

The dynamical equations are

$$\partial_t C_x(t,t') = C_{xv}(t',t) \,;$$
(73)

$$m\partial_t C_v(t,t') = -C_v(t,t') + \int_0^t dt'' R_{x|v}(t,t'')Q''[C_x(t,t'')]C_{xv}(t'',t') +$$
$$+ \int_0^{t'} Q'[C_x(t,t'')]R_v(t',t'') - \mu C_x(t,t')\left(C_x(t,t') - 1\right);$$
(74)

$$\partial_t C_{xv}(t,t') = C_v(t,t') \,;$$
(75)

$$m\partial_{t'} C_{xv}(t,t') = -C_{xv}(t,t') + \int_0^{t'} dt'' R_{x|v}(t',t'')Q''[C_x(t',t'')]C_x(t,t'') +$$
$$+ \int_0^t Q'[C_x(t',t'')]R_{x|v}(t,t'') - \mu C_{xv}(t,t')\left(C_x(t,t') - 1\right);$$
(76)

$$m\partial_t R_v(t,t') = \delta(t-t') - R_v(t,t') + \int_{t'}^t dt'' Q''[C(t,t'')]R_{x|v}(t,t'')R_{x|v}(t'',t') +$$
$$- \mu R_{x|v}(t,t')\left(C_x(t,t') - 1\right);$$
(77)

$$\partial_t R_{x|v}(t,t') = R_v(t,t').$$
(78)

and $\mu(t) = C_{xv}(t, t)$.

The initial conditions are :

$$C_x(t, t) = 1 \; ; \tag{79}$$
$$C_v(t = 0, t = 0) = 0 \; ; \tag{80}$$
$$C_{xv}(t = 0, t = 0) = 0 \; ; \tag{81}$$
$$R_v(t^+, t^+) = 1/m \; ; \tag{82}$$
$$R_{x|v}(t, t) = 0 \; . \tag{83}$$

Where Eq. (79) comes from the spherical constraint; Eqs. (80,81) come from the initialization with no kinetic energy $\boldsymbol{v}(0) = \boldsymbol{0}$; Eqs. (82) and (83) come from Eqs. (69) and (70) (respectively) after deriving by $\Xi(t')$, integrating on $t$ in $[t - h; t + h]$ (with $h \to 0$) and taking $t' \to t$. The equations shown in the main text are the discrete equivalent of the ones just obtained.

## D  Hard spherical constraint

It is also possible to consider a hard spherical constraint, which is the situation typically considered in the physics literature [32, 36]. A massive dynamics was already considered in [52] but their derivation was in the underdamped regime where the total energy is conserved. Using that approach the conservation of the energy was key. Unfortunately the energy is not conserved in general, and in particular in the case of optimization where we aim to go down in energy in order to find a minimum.

For reference sake we consider massive gradient flow, the same considerations apply straightforwardly to Nesterov acceleration. Let us write the dynamics in this case splitting the system in two ODEs

$$m\dot{\boldsymbol{v}}(t) = -\boldsymbol{v}(t) - \nabla \mathcal{L}[\boldsymbol{x}(t)], \tag{84}$$
$$\dot{\boldsymbol{x}}(t) = \boldsymbol{v}(t) - \mu(t)\boldsymbol{x}(t). \tag{85}$$

The last term in the second line constraints the dynamics to move in the sphere by removing the terms of the velocity that move orthogonally to the sphere. Therefore the term $\mu(t)$ is given by the projection of the velocity in the direction that is tangent to the sphere $\sum_j x_j(t)v_j(t)/N$.

We can then follow the usual techniques, e.g. section C, and obtain

$$\partial_t C_x(t, t') = -\mu(t)C_x(t, t') + C_{xv}(t', t) \, , \tag{86}$$

$$m\partial_t C_v(t, t') = -C_v(t, t') + \int_0^t dt'' R_{x|v}(t, t'')Q''[C_x(t, t'')]C_{xv}(t'', t') + \int_0^{t'} Q'[C_x(t, t'')]R_v(t', t'') \, , \tag{87}$$

$$\partial_t C_{xv}(t, t') = -\mu(t)C_{xv}(t, t') + C_v(t, t') \, , \tag{88}$$

$$m\partial_{t'} C_{xv}(t, t') = -C_{xv}(t, t') + \int_0^{t'} dt'' R_{x|v}(t', t'')Q''[C_x(t', t'')]C_x(t, t'') + \int_0^t Q'[C_x(t', t'')]R_{x|v}(t, t'') \, , \tag{89}$$

$$m\partial_t R_v(t, t') = \delta(t - t') - R_v(t, t') + \int_{t'}^t dt'' Q''[C(t, t'')]R_{x|v}(t, t'')R_{x|v}(t'', t') \, , \tag{90}$$

$$\partial_t R_{x|v}(t, t') = -\mu(t)R_{x|v}(t, t') + R_v(t, t'); \tag{91}$$

with $\mu(t) = C_{xv}(t, t)$ and initial conditions $C_x(t, t) = 1$, $C_v(0, 0) = 0$, $C_{xv}(0, 0) = 0$, $R_v(t^+, t) = \frac{1}{m}$, $R_{x|v}(t, t) = 0$.