# OpenReview forum: "Analytical Study of Momentum-Based Acceleration Methods in Paradigmatic High-Dimensional Non-Convex Problems"
_NeurIPS.cc/2021/Conference — NeurIPS 2021 Poster_

### Official Review · Reviewer_JN9a · 2021-07-11

**Rating:** 6
**Confidence:** 4

**Summary:**

This work studies two types of high-dimensional non-convex optimization problems: (1) the mixed p-spin model; (2) the spiked matrix tensor model. Authors studied a high-dimensional, mean-field type limit of momentum-accelerated optimization dynamics solving these two types of non-convex problems. Mean-field type dynamical equations for Polyak's heavy-ball and Nesterov's method are derived.

**Limitations And Societal Impact:**

adequately addressed.

**Main Review:**

The idea of considering a high-dimensional limit of optimization dynamics using mean-field methodology is novel and interesting. It would be great if authors can derive more about the gap between the mean-field limit and the high (but finite) dimensional case. Also, it would be interesting to point out why mean-field equations are useful in understanding these optimization dynamics, such as  its connection with deep learning.

**Time Spent Reviewing:**

2 hours

---

> ### Author Response · Authors · 2021-08-05
> **Reply**
>
> We thank the reviewer for reviewing our work. The first point of the reviewer is very relevant but unfortunately for the moment analyzing finite-size/finite-dimensional corrections to those equations is not possible. This problem is not only relevant for the high-d inference problem, but it is essential also in glass physics itself where the same models appear to be paradigmatic. Even in that case, there have been several unsuccessful attempts to characterize the finite-size effects which remain a hard open problem. In our work, we provided a numerical analysis of the finite-size effects in the spiked model (Section B of SM), but we can not give an analytical answer to that.
>
> About the second point of the reviewer, we do not claim to make a contribution to deep learning. Our study aims at analyzing the dynamics of momentum-based algorithms in a very controlled setting of a high-d non-convex inference problem. In this setting, we can control the degree of non-convexity (i.e. decreasing \Delta_p) and see the effects on the dynamics of the algorithms (and even in this simple scenario the effects are non-trivial). A step forward towards the understanding of neural networks is the study of 1-layer nets, which is achievable with a combination of our techniques with the one of [Mignacco et al. 2020] or simple inference problem seen from the learning point of view such as the phase retrieval problem, see [Mignacco et al. 2021].

---

### Official Review · Reviewer_AYp2 · 2021-07-13

**Rating:** 3
**Confidence:** 5

**Summary:**

The authors try to understand analytically momentum-based accelerated methods in non-convex setting.

**Limitations And Societal Impact:**

The authors should propose some efficient proofs, not the descriptions. I hope the authors can provide more information to make sure their claim is more convincible.

**Main Review:**

The  originality is very good for us to understand analytically momentum-based accelerated methods in non-convex setting and propose a new (spiked) matrix-tensor model. But the quality and clarity is not good, I have never seen how the new model is better than others and how the model is related to momentum-based accelerated methods  in non-convex setting.  Moreover, I have never seen any significance in this manuscript.

**Time Spent Reviewing:**

35

---

> ### Author Response · Authors · 2021-08-05
> **Reply**
>
> We are sorry that the goal of our paper was not clear to the reviewer. The exact characterization of the dynamics in non-convex high-dimensional settings is very rare and hard to obtain; this is one of the few cases where closed equations are obtainable. The contributions of our work are: 1. show how to extend DMFT techniques to momentum-based algorithms; 2. show that (at least in the analyzed inference model) momentum-based algorithms do not improve the recovery threshold but just lead to faster convergence to the ground truth of the problem.
>     The model we analyze is not new from the physics standpoint but its twist as a high-d inference problem was introduced in the ML community in [Mannelli et al. ICML and NeurIPS 2019].
>
> We disagree about the significance. The other reviewers found our paper relevant for the community, and similar analyses and models were deemed to be relevant for the community since they have appeared already in ICML 2019 and NeurIPS 2019.
>
> The method is not rigorous but finds a close match with numerical simulations. Please refer to the answer to reviewer 9jn4 for how the result can be made rigorous. There is no obligation of providing rigorous proofs in NeurIPS papers, and indeed a large number of theoretical results do not.

---

### Official Review · Reviewer_KU8M · 2021-07-18

**Rating:** 5
**Confidence:** 4

**Summary:**

The paper derives a dynamical model for a few numerical solvers for the spiked matrix-tensor model, and analyzes the performance of the accelerated methods using the model. The main contribution is the derivation of the analytic form of the model.

**Limitations And Societal Impact:**

Yes.

**Main Review:**

The main strength of this paper is the derivation of closed-form dynamical models for different solvers, as well as the performance analysis based on the model. The results appear to be new.

The are some issues with the presentation that will benefit from clarification:

- Both the abstract and the end of the first section claim that the accelerated method improves the convergence speed. Is this conclusion derived from the proposed model? Or is this already known in previous work? Given that this is a non-convex problem and the gradient descent solver uses a fixed step size parameter, it is likely that the convergence/acceleration of these methods relies on specific parameter values. So under what settings are these models analyzed? Were you already assuming that all the considered methods converge/accelerate under the given parameters?

- The models were proposed based on a relaxed formulation of the problem. i.e., the hard constraints have been replaced with a soft constraint term with a specific weight parameter mu. The choice of mu will affect the accuracy of the resulting solution (assuming the methods converge). Unless mu approaches infinity, the solution will deviate from the actual solution to the original problem. So how should we interpret the performance analysis? Shall we consider it as performance analysis for the approximate solver which may not indicate the actual performance if we want to look for an accurate solution? One of the conclusions is that the accelerated methods do not alter the algorithmic threshold. Is this reliant on the choice of mu? Can you prove that this behaviour persists no matter how you choose mu?

- At the end of the paper, it is claimed that "the same techniques apply to more recent methods such as quasi-hyperbolic momentum and proportional integral-derivative control algorithm". Looking at this paper only, it is not very clear how the derivation of these analytical model can be carried out for arbitrary methods. (In fact the derivation process for the proposed models were not clear either.) It appears that each new model may need a specific set of mathematical derivations. Can you clarify if there is a common way to analytically analyze these models?

- Line 128 claims that Nesterov acceleration has "quadratic convergence rate". Were you suggesting that Nestertov acceleration has the same convergence rate as Newton's method (which is well known to be quadratically convergent)?

**Time Spent Reviewing:**

5

---

> ### Author Response · Authors · 2021-08-05
> **Reply**
>
>  We thank the reviewer for the report.
>  - We apologize for the confusion. The improved convergence speed is just an empirical observation that we make after deriving the dynamical mean-field equations and integrating them numerically. Our technique does not allow us to compute the convergence rate of the algorithms. In our analysis, we considered several values of the parameters and found consistently a dynamical speed up when the parameters were set to their optimal value. It is still possible that for some $\beta$ small HB is outperformed by GD, but we did not study this case in particular and made not an extensive analysis in this direction, we will add these tests in the revised version.
> - The parameter \mu was introduced in the hope that the additional degree of freedom could lead to an improvement over the standard formulation, but this is not the case. The behaviour persists independently of \mu. Unfortunately, this can not be rigorously proven but is possible to repeat the analysis described in section 5 and observe qualitatively the same result.
> - Yes, there is a common way to derive the mean-field equations. Given an algorithm like quasi-hyperbolic momentum or PID, the idea consists in assuming that the system in dimension $N$ and dimension $(N+1)$ behave in the same way except for a small perturbation induced by the additional degree of freedom. The effect of the additional degree of freedom can be treated in perturbation theory and the only relevant correction is the first one (so that linear response theory is exact in the large dimensional limit), while higher-order corrections give subleading contributions in $1/N$ to the final result. In practice the steps to follow are described in Section 4, namely: writing the cavity equations Eqs. (28-30), expanding around the perturbation induced by the additional degree of freedom, recognizing the order parameters. This comment is very relevant to us since one of the main goals of this paper is to provide a method for analyzing such problems. We will work on improving the clarity (as was also suggested by reviewer 9jn4).
> - We apologize for the confusion, that is not our result, we were just reporting results from the literature (ref. [9]). We will correct this statement in the revisited version of the manuscript.

---

> > ### Comment · Area_Chair_jfdd · 2021-09-19
> > **Quadratic convergence**
> >
> > This mismatch may be due to the fact that quadratic convergence is sometimes used for sublinearly-convergent algorithms whose error goes down as $\frac{1}{T^2}$, such as Nesterov's method for smooth convex functions.

---

### Official Review · Reviewer_9jn4 · 2021-07-20

**Rating:** 7
**Confidence:** 4

**Summary:**

The authors study the dynamics of accelerated gradient methods applied to two prototypical, non-convex problems: recovering the ground state of a mixed p-spin model, and recovering a planted signal from a spiked matrix-tensor model. The two models are defined according to the following loss function

$$
\mathcal{L}(x) = - \eta_p(\Delta_p) \cdot \sum_{i_1, \ldots, i_p}^N T_{i_1, \ldots, i_p} \cdot x_{i_1} \ldots x_{i_p} - \eta_2(\Delta_2) \cdot \sum_{i,j}^N Y_{i,j} \cdot x_{i} x_{j}
$$

where $T \in (\mathbb{R}^N)^{\otimes p}$ is a tensor, $Y \in \mathbb{R}^{N \times N}$ is a matrix and $\eta_p(\Delta_p)$ and $\eta_2(\Delta_2)$ are normalization constants dependent on $\Delta_p$ and $\Delta_2$ respectively. When $T$ and $Y$ possess entries that are i.i.d Gaussians with mean 0 and variance $\Delta_p$ and $\Delta_2$ respectively, we have the mixed $p$-spin model. When $T$ and $Y$ are given by a Gaussian perturbation of vector $x^*$ on the $\sqrt{N}$ radius $N$-dimensional sphere, we have the spiked matrix-tensor model. Finding the ground state for mixed $p$-spin, and recovering the planted signal from the spiked tensor model then correspond to finding $x$ which minimizes loss $\mathcal{L}$. Note in the latter case that $x$ which minimizes loss will also have a large overlap with $x^*$ hence why one considers this problem as recovering a planted signal.

This work uses the cavity method to derive mean dynamics for iterates produced by accelerated gradient methods. They construct a sequence of equations that track correlation between iterates maintained by Nesterov’s acceleration and Polyak’s Heavy Ball Momentum. These equations are parameterized by different time steps of the algorithm; for example, the equation for $C_x[t, t’]$ corresponds to the overlap of $x$ at time $t$ with that at time $t’$.

The dynamic mean field equations can be used to heuristically determine computational thresholds for statistical signal recovery in the following sense. For an instance of finding a planted signal from a spiked matrix-tensor model given by samples of $T$ and $Y$, consider the instance’s signal-to-noise ratio $\frac{\Delta_p}{\Delta_2}$. For a fixed value of $\Delta_p$ and as one increases $\Delta_2$, it becomes increasingly difficult to recover the planted spike $x^*$. At a certain value of $\Delta_2$, say $\delta^*$, an iterative method solving the instance may exhibit a phase transition: for $\Delta_2 < \delta^*$ the method will find $x$ that has large overlap with $x^*$, while for $\Delta_2 > \delta^*$, the method only converges to $x$ that has vanishingly small (w.r.t. $N$) overlap with $x^*$.

The authors use the mean dynamic equations to compute statistical recovery thresholds for Nesterov acceleration and Heavy Ball momentum. The cavity method calculations are heuristic, and so the authors support their estimates with experiments that suggest the true threshold for these two methods are very close to their estimates. The results also suggest that the threshold for accelerated methods align closely with the threshold achieved by vanilla gradient descent. As a separate results, the authors demonstrate an equivalence between the dynamics of Polyak’s Heavy Ball Momentum and the dynamics of a massive particle under the effect of a certain potential field.

**Limitations And Societal Impact:**

The authors have reasonably addressed these concerns.

**Main Review:**

The primary contribution of this paper that I see is its estimation of a computational threshold when Nesterov's acceleration and Polyak's Heavy Ball momentum are used to recover a planted signal from a spiked matrix-tensor model. The methodology required to compute the dynamic equations and subsequently estimate the threshold are non-trivial and interesting. Even though this use of the Cavity method implies that the calculations are non-rigorous, experiments demonstrate that the estimates align closely with the true computational threshold. One might hope then to use the techniques of (Mannelli et. al., 2019) to rigorously establish this threshold, or alternative methods of establishing the rigor of the cavity method (Bapst et. al., 2016), (Coja-Oghlan & Panagiotou, 2016), (Ding et. al., 2015).

The observation that the computational threshold for accelerated methods align closely with the threshold achieved by “vanilla” gradient-flow algorithms is surprising. Accelerated methods can be viewed as a coupling between the “primal” method of gradient descent, and the “dual” method of mirror descent. Mirror descent has classically been employed in certain combinatorial optimization problems to efficiently construct “certificates” that establish that the target solution cannot exist in a region of the search space. A priori, one might imagine that accelerated methods then recover the planted signal for a broader range of parameters when compared to gradient descent because, simultaneous to searching for the planted signal, accelerated methods also implicitly maintain a certificate regarding where not to search. These experiments suggest that the aforementioned intuition is likely either misguided, or not enough to supersede the gradient descent computational threshold.

A few broad comments.
- Further discussion regarding rigorously establishing the calculations presented in this work is warranted. What prevents making this application of the Cavity method rigorous? Separately, is there a significant roadblock towards extending the results of Mannelli et. al. to encompass accelerated methods?
- Certain approximations are made within the calculations without much warning (c.f. equations 31, 35). These are standard approximations employed in spin glass calculations; however they also increase the inaccessibility of such topics to those who may not have extensive background. When an approximation is made, can there be a brief exposition (perhaps even one sentence) explanation motivating its use?
- The paper lists extending their techniques to other models such as “1-layer feedforward networks, and recurrent networks” as future work. However, broader discussion regarding why one might wish to study statistical recovery thresholds for different models and optimization methods is warranted. For example, does this give some algorithmic insight into designing methods that would recover planted signal for a larger range of parameters?

Other minor comments
- On line 118 $C_{xy}[t, t’]$ as written doesn’t depend on $t’$
- On line 119, please provide a definition for the first time $\delta$ is used.
- On line 119, define what local field is (or specify the dimensions of this object).
- On equation 11, the equation is missing a $t’$.

**Time Spent Reviewing:**

20

---

> ### Author Response · Authors · 2021-08-05
> **Reply**
>
> We thank the reviewer for the thorough summary and for the interesting connection with mirror descent.
>
> - We will comment on how to make the results rigorous. The relevant works on that side are [Ben Arous et al 2006, Dembo & Subag 2019] where the authors provided upper and lower bound to the resulting dynamical equations of gradient flow in the mixed p-spin. Making our results rigorous would be an extension of those works. In particular, a standard approach is to follow a path integral description of the dynamics and prove a large deviation principle for the infinite-dimensional limit. Our dynamical equations emerge directly as saddle point equations.
> - We thank the reviewer for remarking on the steps that were unclear. The idea in the Eqs. (31-35) is to treat the effect of an additional variable by linear response theory. This provides the exact results because the corrections to linear response theory are subleading in the infinite-dimensional limit. We will make add a comment on this in the revisited version of the manuscript.
> - We can extend this work to different models and problems. One can either consider more complex high-d inference problems, as an example, one can repeat our analysis for phase retrieval, or consider simple learning problems. In both those cases, the (stochastic) gradient descent dynamics has been recently analyzed in [Mignacco et al. NeurIPS 2020, Mignacco et al. 2021]. We believe that extending our approach to those settings is doable on the same lines we have developed in this work. Indeed those extensions are anyway based on linear response theory which is the main tool that makes the dynamics (with momentum or not) exactly soluble. We do not foresee insight into designing new algorithms, our motivation is mostly theoretical, but this can not be excluded a priori. Indeed with the scope of understanding SGD in 1-layer networks [Mignacco et al. 2020] through the main techniques we used in the present work, a new algorithm was introduced (persistent-SGD) that appears to out-perform standard SGD [Mignacco et al. 2021].
> - We thank the reviewer for the technical comments, we will fully address them in the revised version of the manuscript.

---

### Decision · Program_Chairs · 2021-09-27

**Decision:**

Accept (Poster)

**Comment:**

The reviewers and AC agree that the idea of considering a high-dimensional limit of optimization dynamics using mean-field methodology to heuristically derive computational thresholds for important statistical recovery methods is interesting and may be conducive towards stronger theoretical results. The experiments demonstrating the accuracy of these predictions are very helpful in supporting the mathematical approach. A number of useful suggestions on presentation were made by the reviewers. The authors should make sure to incorporate these in their final version.